# Elicitation with Bacillus QV15 reveals a pivotal role of F3H on flavonoid metabolism improving adaptation to biotic stress in blackberry

Enrique Gutiérrez-Albanchez[1]*, Ana Gradillas[2], Antonia García[2], Ana García-Villaraco[1], F. Javier Gutierrez-Mañero[1], Beatriz Ramos-Solano[1]*

1 Department of Pharmaceutical and Health Sciences, Facultad de Farmacia, Universidad San Pablo-CEU Universities, Boadilla del Monte, Madrid, Spain, 2 Department of Chemistry and Biochemistry, Facultad de Farmacia, Universidad San Pablo-CEU Universities, Boadilla del Monte, Madrid, Spain

* bramsol@ceu.es (BRS); enriquegutierrez2013@gmail.com (EGA)

## Abstract

The aim of this study is to determine the involvement of the flavonol-anthocyanin pathway on plant adaptation to biotic stress using the *B.amyloliquefaciens* QV15 to trigger blackberry metabolism and identify target genes to improve plant fitness and fruit quality. To achieve this goal, field-grown blackberries were root-inoculated with QV15 along its growth cycle. At fruiting, a transcriptomic analysis by RNA-Seq was performed on leaves and fruits of treated and non-treated field-grown blackberries after a sustained mildew outbreak; expression of the regulating and core genes of the Flavonol-Anthocyanin pathway were analysed by qPCR and metabolomic profiles by UHPLC/ESI-qTOF-MS; plant protection was found to be up to 88%. Overexpression of step-controlling genes in leaves and fruits, associated to lower concentration of flavonols and anthocyanins in QV15-treated plants, together with a higher protection suggest a phytoanticipin role for flavonols in blackberry; kempferol-3-*O*-rutinoside concentration was strikingly high. Overexpression of *RuF3H* (Flavonol-3-hidroxy-lase) suggests a pivotal role in the coordination of committing steps in this pathway, control-ling carbon flux towards the different sinks. Furthermore, this C demand is supported by an activation of the photosynthetic machinery, and boosted by a coordinated control of ROS into a sub-lethal range, and associated to enhanced protection to biotic stress.

## Introduction

*Rubus* cv. Loch Ness is a plant that belongs to a large group of plants with beneficial properties for human health known as berries. This group is characterized for the high amount of secondary metabolites (flavonoids among others) present in their fruits, and in leaves [1–3]; benefits for human health relay on flavonoids to a great extent [4–5].

Plants have successfully colonized all environments of our planet, thanks to their ability to develop a plant-specialized metabolism as a part of their evolutionary process, which enables

**Data Availability Statement:** All relevant data are within the manuscript and its Supporting Information files.

**Funding:** This work was supported by grants from the Spanish Ministerio de Economía y Competitividad for projects AGL2013-45189 (BRS, JGM), CTQ2014-55279-R (AG) and grant reference BES-2014-0769990 to EGA. The funders had no role in study design, data collection and analysis, decision to publish, or preparation of the manuscript.

**Competing interests:** No authors have competing interests.

them to adapt to the continuous changing conditions along their lifetime [6]. Plant secondary metabolism confers plasticity to plants so that they are able to adapt to changing environmental conditions, usually adverse conditions, ensuring plant survival [7–8]. Hence, this metabolism is sensitive to different factors among which are biotic agents, like beneficial or harmful microorganisms, which can be used to trigger plant metabolism. Therefore, beneficial microorganisms constitute a biotechnological tool to improve plant fitness and enhance secondary metabolites contents in plant organs [9–11]. More precisely, the use of beneficial bacteria to trigger secondary metabolism involved in plant defense is gaining a lot of interest and there is increasing evidence of their effectiveness under controlled and field conditions to support their effects [12]. Furthermore, elicitation can be used as a tool to identify target genes to be edited by CRISPR/Cas9 with the aim to improve plant fitness and/or food quality [13].

Plant Growth Promoting Bacteria (PGPB) are beneficial strains naturally present in the rhizosphere of plants contributing to plant health. As it has been demonstrated certain strains trigger expression of some plant genes, defense related genes among others, enhancing plant defense metabolism; so when the pathogen tries to invade the plant, it is already prepared and not dramatically infected [14, 15]. Therefore, PGPB appear as an alternative to chemical pesticides as well as tools to study plant metabolism. As pests in the agricultural systems are an important threat because they reduce plant yield and fruit quality, with the consequent economic losses, pest control is an unquestionable challenge for agriculture and to achieve food security, a term which refers to "food availability, in sufficient quantities with proper amount of nutrients and on a consistent basis". Hence, finding effective biological agents is a challenge, and unraveling plant changes upon delivery of the biological will set the bases for a successful agronomic management.

The present study focuses on flavonoid metabolism, as it is highly expressed in blackberry, and both leaves and fruits contain high flavonoid concentrations [16]. Flavonoids belong to a metabolic network that mediates on plant adaptation to environmental stress Flavonoids are a ubiquitous group of secondary metabolites key for adaptation and survival on earth life [17], participating in many different processes of plant physiology [4, 6, 18–23]. Hence, a deeper knowledge of flavonoid metabolism and key enzymes controlling relevant branching points will allow us to manipulate plant metabolism in our benefit, for example in agriculture. Upon biotic stress challenge, they may play a different role in defense, as they can either be defensive molecules by themselves, behaving as phytoalexins, or they may be accumulated as phytoanticipins and transformed into the real phytoalexins upon pathogen challenge [24–25].

Based on this background, and using *B.amyloliquefaciens* QV15 as a tool to trigger plant metabolism, the present study reports the systemic effects of root inoculated bacteria on blackberry leaves and fruits at the transcriptomic and metabolomic level, focusing on the flavonol-anthocyanin biosynthetic pathway (Fig 1). The aim of this study was (i) to study the effects of elicitation with a beneficial biotic agent on blackberry leaf and fruit metabolism, (ii) to determine the involvement of the flavonol-anthocyanin pathway on plant adaptation to biotic stress. To achieve these objectives, a transcriptomic analysis by RNA-Seq was performed, and qPCR expression of the regulating and core genes of the Flavonol-Anthocyanin pathway and metabolomic changes by UHPLC/ESI-qTOF-MS analysis on inoculated and non-inoculated field-grown blackberries at fruiting, after a sustained mildew outbreak were determined.

## Materials and methods

### Bacterial strain

*Bacillus amyloliquefaciens* QV15 (CECT 9371) is a gram positive sporulated bacilli, isolated from the rhizosphere of *Pinus pinea* [26]. Able to produce siderophores and to stimulate pine

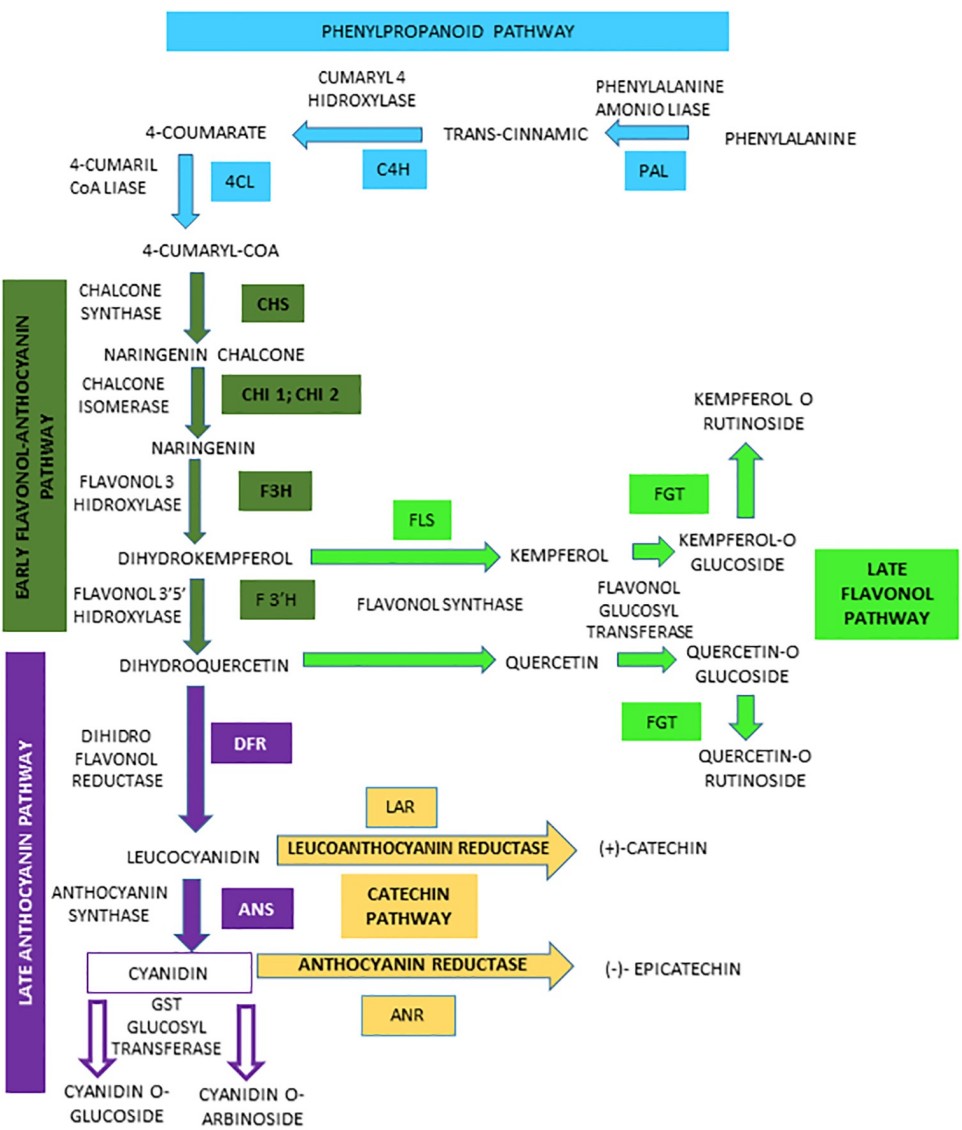

**Fig 1. Biosynthesis of anthocyanins, flavonols and catechins via the flavonoid pathway in *Rubus* cv. Loch Ness.**
Phenylalanine ammonio-lyase (RuPAL1 and RuPAL2), Cinammate-4-hydroxylase (RuC4H), 4-coumaryl-CoA ligase (Ru4CL), Chalcone synthase (RuCHS), Chalcone Isomerase1 (RuCHI1), Chalcone Isomerase2 (RuCHI2), Flavonol-3-hydroxylase (RuF3H), Flavonoid 3´5´hydroxylase (RuF3´5´H), Flavonoid 3´hydroxylase (RuF3´H), Flavonol synthase (RuFLS), Leucoanthocyanidin reductase (RuLAR), Anthocyanidin reductase (RuANR), Dehydroflavonol reductase (RuDFR), Anthocyanidin synthase (RuANS), Flavonol and Anthocyanidin Glycosiltransferases (RuFGT and RuAGT).

growth [27], it is also able to enhance defence against *Pseudomonas syringae* (DC3000) and to protect against abiotic stress (NaCl 60 mM) [28].

Bacterial strain was maintained at -80 ˚C in nutrient broth with 20% glycerol. Inoculum was prepared by streaking strains from -80 ˚C onto plate count agar (PCA) plates, incubating plates at 28 ˚C for 24 h. After that, QV15 was transferred to Luria Broth liquid media (LB) that was grown under shaking (1000 rpm) at 28 ˚C for 24 hours to obtain a $2x10^9$ cfu/mL inoculum.

## Plant materials and experimental set up

*Rubus* cv. Loch Ness is a high yielding tetraploid (4n = 28) blackberry, and one of the most widely cultivated varieties. In southwest Spain, blackberries are produced under "winter cycle" involving an artificial cold period in order to induce flowering upon transplant to greenhouses. Blackberry cycle has three stages: vegetative, flowering and flowering-fruiting; the duration of these stages is variable depending on the transplant moment, and each stage approximately accounts for one third of the plant's life. In this experiment, plants were transplanted at the end of September 2014, flowering took place in November and maximum fruiting in January 2015.

The *Rubus cv*. Loch Ness plants used in this study were kindly provided by Agricola El Bosque S.L. (Lucena del Puerto, Huelva, Spain). Plants and greenhouses were handled according to regular agricultural practices [29]. A total of 360 plants were in the trial, arranged in six greenhouses; each greenhouse had two lines with 60 plants each, being each line one replicate with 60 repetitions; 3 lines were inoculated, and 3 lines were left as non-inoculated controls. QV15 was root inoculated every 15 days during the whole plant cycle, with 0.5 L of inoculum at $10^7$ cfu/ml per plant.

Number of flowers per square meter at flowering, and accumulated fruit production, were recorded. A natural Mildew outbreak took place from November till harvest, and disease incidence was recorded by visual evaluation of the surface affected carried out by 3 independent expert observers. Relative disease index has been presented as surface affected on QV15 treated plants relative to that of controls (%). At fruiting, leaves and fruits were sampled, immediately frozen in liquid nitrogen, and brought to the lab. Three replicates were taken, being each one constituted from plant material of 60 plants; leaves were randomly sampled and pooled constituting one replicate as well as red fruits. Samples were powdered with liquid nitrogen and two aliquots were made, one for RNA extraction for transcriptome analysis, qPCR of core and regulatory genes of the flavonol-anthocyanin pathway and the other aliquot for pigments (chlorophylls) and bioactives (phenols, flavonols, anthocyanins) determination, by colorimetry and UHPLC/ESI-qTOF-MS.

## RNA-Seq

**RNA extraction, quality control and Library preparation.** Total RNA was isolated from each leaf and fruit replicate with Plant/Fungi Total RNA Purification kit [30] (NORGENTM) (DNase treatment included). A reverse transcription followed by a RT-qPCR and RNA-Seq were performed. Thirty μl of RNA samples were passed through quality control with Nanodrop™ and Experion™, after that, total RNA meeting quality criteria was sent to Sistemas Genómicos for sequencing. A total of three libraries were done for each organ.

**RNA library assembly.** Ribosomal RNA removal was performed with the Ribo-Zero rRNA removal kit. Generation of libraries was performed with the TruSeq Stranded Total RNA library Prep kit following manufacturer's recommendations. Two μg of total RNA (RIN>9) libraries were sequenced using a HiSeq2500 instrument (Illumina Inc, San Diego, CA, USA). Sequencing readings were paired-end with a length of 101bp, and reading was performed in 6 samples. The estimated coverage was around 52 million reads per sample (1 lane). Library generation and RNA sequencing was done at Sistemas Genómicos S.L. (Valencia, Spain) following manufacturer´s instructions.

**RNA transcriptomic analysis: Alignment, annotation, classification of genes and differential expression.** The quality control of the raw data was performed using the FastQC v0.11.4 tool. For alignment, the raw paired-end reads (c-DNA) were mapped against the *Rubus occidentalis* genome v1.1 provided by GDR database using Tophat2 2.1.0 algorithm

[31]; the genome used is from the same genus, but different species. Insufficient quality reads (phred score<5) were eliminated using Samtools 1.2 [32] and Picard Tools 2.12.1. In this step, the GC distribution (i.e. the proportion of guanine and cytosine bp along the reads) was assessed expecting a desired distribution between 40–60%. Second, distributions of duplicates (quality of sequencing indicator) were evaluated to confirm that our sequencing contained small proportion of duplicates.

Genes were then used for BLAST searches and annotation against NCBI Nr protein database (NCBI non-redundant sequence database). Blast was performed by CloudBlast, using Blastx-fast Blast Program, Non-redundant protein sequences (nr) from 17.01.2016 as Blast DB (Blast database), with a blast expectation value of $1 \times 10^{-10}$, a word size of 6, and a HSP length cutoff of 33. Gene sequences were then aligned by BLASTX to protein databases (Swiss-Prot, KEGG), retrieving proteins with the highest sequence similarity with the given genes along with the functional annotations for their proteins. When conflict appeared from different databases, a priority order of Nr, Swiss-Prot, and KEGG was followed. For everything previously described, Blas2GO® was used [33] as well as to obtain GO annotations for the genes, and for mapping and annotation (mapping is used to look for associated GO terms to the Hit Blast, and annotation selected from this GOs those with good statistical support). A cutoff of 55, and a GO weight of 5 were used for annotation.

Expression levels were calculated using the HiTSeq [34]. This method employs unique reads for the estimation of gene expression and filters the multimapped reads. Differential expression analysis between conditions was assessed using DESeq2 [35].

Finally, criteria for identifying differentially expressed genes were a P value adjusted by FDR≤0,05 [36] and a fold change of at least 1.2. The DEG analysis between groups was done using statistical packages designed by Python and R. Using DESeq2 algorithm [34], applying a differential negative binomial distribution for the statistics significance [31–35], enabled us to identify genes differentially expressed. We considered as differently expressed genes those with a FC value below −1.2 or higher than 1.2 and with P value (Padj) corrected by FDR≤0,05 to avoid identification of false positives across the differential expression data. The FC is the log-ratio of a gene's or a transcript's expression values in two different conditions. The FC is used to confirm the significance of the differential expression between the different samples [FC ≥ 1 (overexpression in treatment), or FC ≤ −1 (overexpression in controls)].

## Photosynthetic pigments (chlorophylls and carotenoids) extraction and quantification in leaves

Chlorophylls were isolated from each replicate. One hundred mg of powdered leaves were dissolved in 3 mL of acetone 80%, vortexed and centrifuged for 5 min at 10000 r.p.m (Hermle Z233 M-2). Absorbance was measured at 645, 662 and 470 nm in a Biomate 5 spectrophotometer. To calculate chlorophyll a, chlorophyll b, and carotenoids, the following formulas were used [37].

- Chl a (mg g$^{-1}$) = [(12.25 x Abs$_{663}$)—(2.79 x Abs$_{647}$)] x V (ml)/ weigh (mg).

- Chl b (mg g$^{-1}$) = [(21.5 x Abs$_{647}$)—(5.1 x Abs$_{663}$)] x V (ml)/ weigh (mg).

- Carotenoids (mg g$^{-1}$) = ((1000 x Abs$_{470}$)–(1.82 x Chl a)- (85.02 x Chl b))/198) x V (ml)/ weigh (mg).

## Bioactive characterization

**Colorimetric quantification.** Two extracts were prepared. One g of powdered leaves from each replicate was mixed with 9 mL of methanol 80% for phenols and flavonols

determination, and one gram with 9 mL of methanol 80% 0.1% HCl for anthocyanins determination; then, samples were sonicated for 10 min and centrifuged for 5 min at 5000 rpm. Supernatants were frozen and lyophilized. The same was done with fruit samples.

Total phenols were quantitatively determined with Folin-Ciocalteu reagent (Sigma. Aldrich, St Louis, MO) by a colorimetric method described in [38], with some modifications. One milliliter of extract was mixed with 0.250 mL of Folin-Ciocalteu 2N (Sigma. Aldrich, St Louis, MO) and 0.75 mL of $Na_2CO_3$ 20% solution. After 30 minutes at room temperature, absorbance was measured at 760 nm with an UV-Visible spectrophotometer (Biomate 5). A gallic acid (Sigma-Aldrich, St Louis, MO) calibration curve was made (r = 0.99). Results are expressed in mg of gallic acid equivalents per g of powdered leaves. The same was done with fruit samples. All samples were measured in triplicate.

Total flavonols were quantitatively determined through the test described in [39]. One milliliter of extract was added to a 10 mL flask with 4 mL of distilled water. After that, 0.3 mL of $NaNO_2$ 5%, and 0.3 mL of $AlCl_3$ 10% were added after 5 minutes. One minute later, 2 mL of NaOH 1 M were added and the mixture was brought to 10 mL with distilled water. The solution was mixed and measured at 510 nm with an UV-Visible spectrophotometer (Biomate 5). A catechin (Sigma-Aldrich, St Louis, MO) calibration curve was made (r = 0.99). Results are expressed as mg of catechin equivalents per g of powdered leaves. The same was done with fruits. All samples were measured in triplicate.

Total anthocyanins were quantitatively determined through the pH differential method described by [40]. Extracts were diluted in pH 1 buffer (0.2 M KCl) and pH 4.5 (1M $CH_3COONa$) in 1:15 proportion. After that, absorbance was measured at 520 and 720 nm respectively, in a UV-Visible spectrophotometer (Biomate 5). A cyanidin-3-glucoside (Extrasynthese Co.™, Geney, France) calibration curve was made (r = 0.99). Results are expressed in cyanidin-3-glucoside equivalents per g of powdered leaves. The same was done with fruits. All samples were measured in triplicate.

**UHPLC/ESI-qTOF-MS phenolics and flavonoids analysis.** Standards and solvents: Phenolic acids including, gallic acid, caffeic acid, ferulic acid and chlorogenic acid were purchased from Sigma (St. Louis, MO, USA) and flavonoids including, kaempferol, kempherol-3-*O*-rutinoside, kempherol-3-*O*-glucoside, quercetin, quercetin-3-*O*-rutinoside, quercetin-3-*O*-glucoside, (+)-catechin, (-)-epicatechin and cyanidin-3-*O*-glucoside, were purchased from Sigma and from Extrasynthese Co. ™ (Geney, France).

The standard solutions (10 ppm) were prepared in methanol. All the solvents, as methanol and acetonitrile (*Honeywell Riedel-de Haen*), were LC-MS grade. Purified water was obtained from Milli-Q Plus™ System from Millipore (Milford, MA, USA). Formic acid was purchased from Aldrich (St. Louis, MO, USA).

*Sample preparation*. Extraction of phenolics was performed as follows: 30 mg of powder were added to 300 μL of methanol. The mixture was vortexed for 2 min, sonicated for 5 min and centrifuged at 3.500 rpm for 5 min at 4 ˚C. The supernatants were then collected and stored at -20 ˚C until use for analysis.

*UHPLC/ESI-qTOF-MS Analysis*. Samples were analyzed on a 1290 Infinity series UHPLC system coupled through an electrospray ionization source (ESI) with Jet Stream technology to a 6550 iFunnel QTOF/MS system (Agilent Technologies, Waldbronn, Germany) as described in [41].

For the separation, a volume of 2 μL was injected in a reversed-phase column (Zorbax Eclipse XDB-C18 4.6 × 50 mm, 1.8 μm, Agilent Technologies) at 40 ˚C. The flow rate was 0.5 mL/min with a mobile phase consisted of solvent A: 0.1% FA, and solvent B: methanol. Gradient elution consisted of 2% B (0–6 min), 2–50% B (6–10 min), 50–95% B (11–18 min), 95% B for 2 min (18–20 min), and returned to starting conditions 2% B in one minute (20–21 min) to finally keep the re-equilibration with a total analysis time of 25 min.

Detector was operated in full scan mode (*m/z* 50 to 2000), at a scan rate of 1 scan/s. Accurate mass measurement was assured through an automated calibrator delivery system that continuously introduced a reference solution, containing masses of *m/z* 121.0509 (purine) and m/z 922.0098 (HP-921) in positive ESI mode; whereas *m/z* 112.9856 (TFA) and m/z 922.009798 (HP-921) in negative ESI mode. The capillary voltage was ±4000 V for positive and negative ionization mode. The source temperature was 225 ˚C. The nebulizer and gas flow rate were 35 psig and 11 L/min respectively, fragmentor voltage to 75V and a radiofrequency voltage in the octopole (OCT RF Vpp) of 750 V.

For the study, MassHunter Workstation Software LC/MS Data Acquisition version B.07.00 (Agilent Technologies) was used for control and acquisition of all data obtained with UHPLC/ESI-qTOF-MS.

For quantification, each standard was injected twice in four different concentrations to build up callibration curves in which sample peak areas were interpolated.

*Data treatment*. UHPLC-MS data processing was performed by MassHunter Qualitative Analysis (Agilent Technologies) Software version B.08.00 using Molecular Feature Extraction (MFE).

**RT-qPCR analysis of the flavonol-anthocyanin pathway.** Retrotranscriptions were performed using iScript tm cDNA Synthesis Kit (Bio-Rad) on a GeneAmp PCR System 2700 (Applied Biosystems) in the following conditions: 5 min 25 ˚C, 30 min 42 ˚C, 5 min 85 ˚C, and hold at 4 ˚C. Amplifications were performed with a MiniOpticon Real Time PCR System (Bio-Rad) in the following conditions: 3 min at 95 ˚C and then 39 cycles consisting of 15 s at 95 ˚C, 30 s at 55 ˚C and 30 s at 72 ˚C, followed by melting curve to check the results. Cycle threshold (Ct) was used to describe expression obtained in the analysis. Standard curves were calculated for each gene, and the efficiency values ranged between 90 and 110%. *HISTONE H3* (*HIS*) was used as reference gene. Results for gene expression were expressed as differential expression by the $2^{-\Delta\Delta Ct}$ method. Control expression is set at 1, therefore only increases above one are considered. Core and regulatory genes were studied, and the primers used for each appear in S3 Table.

## Statistical analysis

To evaluate treatment effects on photosynthetic pigments, bioactive contents and gene expression, one-way ANOVA analysis were performed. When significant differences appeared ($p<0.05$), LSD test (Least significant Difference) from Fisher was used. Statgraphics plus 5.1 for Windows was the program used.

## Results

### Evaluation of plant fitness

There is a significant increase of flowers per square meter in QV15-inoculated plants not associated to an increase in production. The Mildew outbreak started in November and was maintained throughout the plant cycle; controls showed 15% affected surface on average, while QV15 treated plants showed an average 5%. The relative disease index determined at fruiting indicates a rough 88% protection against the natural fungal disease (Table 1).

**Table 1. Plant fitness parametres in controls and QV15 treated plants.**

|  | Flowers/m2 | Production (Kg/plant) | Evolution of disease (% affected surface) | Relative disease index (%) |
|---|---|---|---|---|
| **Control** | 237.95 ± 2.28 (a) | 6.2 ± 0.22 (a) | 15% (b) | 100 ± 1.05 (b) |
| **QV15** | 323.5 ± 1.77 (b) | 6.4 ± 0.09 (a) | 5% (a) | 12.02 ± 0.36 (a) |

Number of flowers per square meter of blackberry plants at flowering. Production (Kilograms per plant). Disease incidence measured as affected leaf surface (%) with Mildew symptoms in blackberry plants from November (21$^{st}$ November) to harvest in february (02/05/2018). Relative disease index expressed as accumulated values of affected leaf surface (%) with Mildew symptoms relative to controls. Different letters denote statistically significant differences according to LSD test (p<0.05).

## RNA-Seq

In a typical experiment of whole transcriptome analysis, the number of mapped lectures to the reference genome is around 50%. In this case, in which the reference genome is *Rubus occidentalis* (v1.0 &Annotation v1 from the database of GDR), the mapping fraction obtained was around 50% (49.97% to 52.43%) (S1 Table).

After sequencing and mapping alignment, normalized and differential expression (Control vs. QV15), a total of 29,126 genes were identified in leaves (S1 Table) and fruits (S2 Table). The heatmap diagram (Fig 2) shows the result of the two-way hierarchical clustering of RNA transcripts and samples; it includes the 50 genes that have the largest coefficient of variation based on FPKM counts in leaves and fruits. Each row represents one gene and each column represents one sample. The color represents the relative expression level of a transcript across all samples. The color scale is shown below: red represents an expression level above the mean; green represents an expression level below the mean.

When leaf samples were compared, the expression pattern showed that 28,586 genes were equally expressed in both treatments (expression without significant differences), 173 genes were significantly overexpressed in leaves of controls, and 367 genes were significantly overexpressed in leaves of QV15-treated plants (Fig 3a). When fruits were compared, expression of 27,866 genes showed non-significant differences, while genes overexpressed in controls accounted for 595, and genes overexpressed in QV15-treated plants accounted for 664 (Fig 3b), being these numbers triple and double than in leaves, respectively.

Overexpressed genes in leaves appear in (S3 File). In controls, most genes are related to the phenylpropanoids-flavonoid pathway, and sugar metabolism. Overexpressed genes in leaves of QV15-treated plants are related to the following aspects: i) an active photosynthesis (mostly related to photosystems I and II), ii) an active regeneration of photosystems including pigments biosynthesis, and iii) an efficient capacity of ROS scavenging, as shown by the high number of transcripts of superoxide dismutase (SOD), and ascorbate peroxidase (APX). It is worth mentioning the high expression of glutathione-S-transferase 2 (GST2) (S2 Table).

Two groups appear among overexpressed genes in control fruits (S4 File): in one hand, a high vacuolar activity is detected together with an active sucrose metabolism, and in the other hand, many transcripts of ubiquitin-protein ligases, serin/theronin kinases and Fbox/FBD/LRRs are detected. In QV15-treated fruits, specialized defense enzymes such as subtilisin, different glucanases and chitinases, and a striking overexpression of GDSL esterase/lipases, a family of proteins that has been related to secondary metabolites synthesis and plant defense appear, among overexpressed genes [42].

## Photosynthetic pigments (chlorophylls and carotenoids)

Chlorophylls and carotenoids were significantly more abundant in leaves of QV15-treated plants (Table 2). Control plants had 0.57 mg g$^{-1}$ Chl a, 0.23 mg g$^{-1}$ Chl b, and 0.38 mg g$^{-1}$ carotenoids, while QV15 treated plants had 0.72 mg g$^{-1}$ Chl a, 0.36 mg g$^{-1}$ Chl b, and 0.42 mg g$^{-1}$

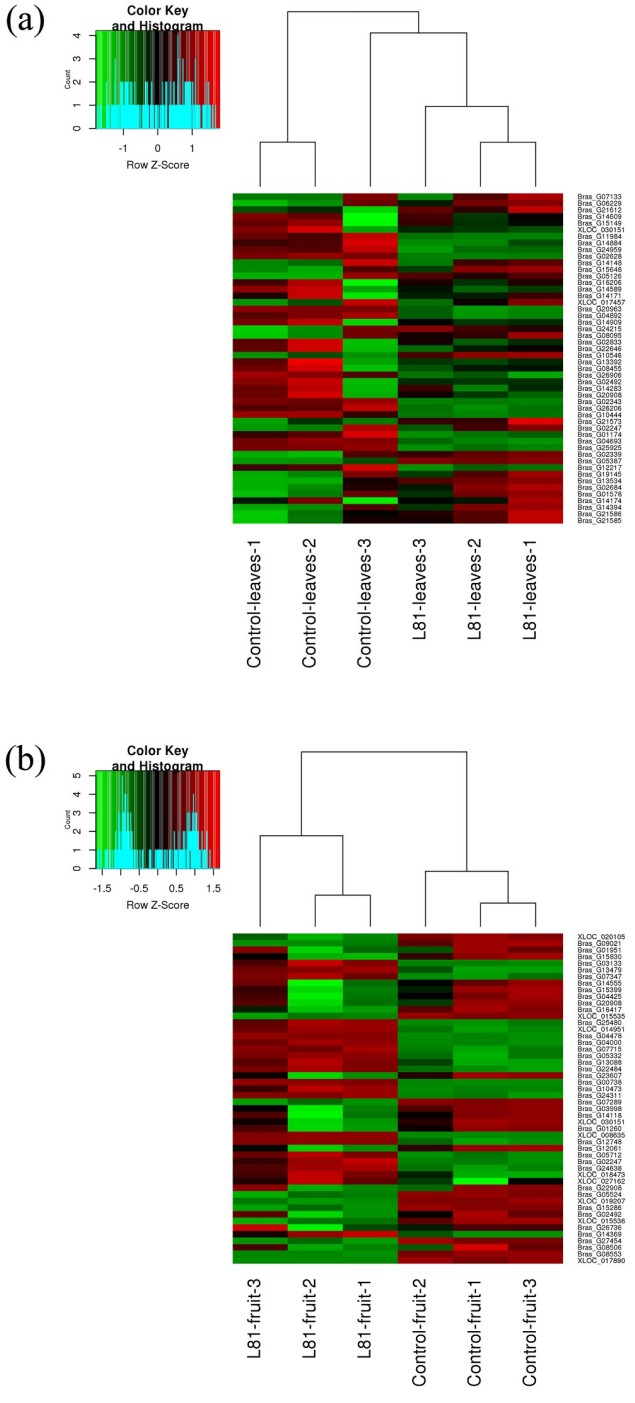

**Fig 2. Heat Map and unsupervised hierarchical clustering by sample; top 50 genes with the largest coefficient of variation based on FPKM counts, a) in leaves b) in fruits.**

carotenoids. These values represent a rough 54% increase in chlorophyll contents, mainly chlorophyll b, and 5% increase in carotenoid content in QV15 treated plants.

Quantification of chlorophyll A, B, and Carotenoids in blackberry leaves. Values are the average of 3 replicates ± SD. Different letters denote statistically significant differences between treatments for each parameter according to LSD test (p<0.05).

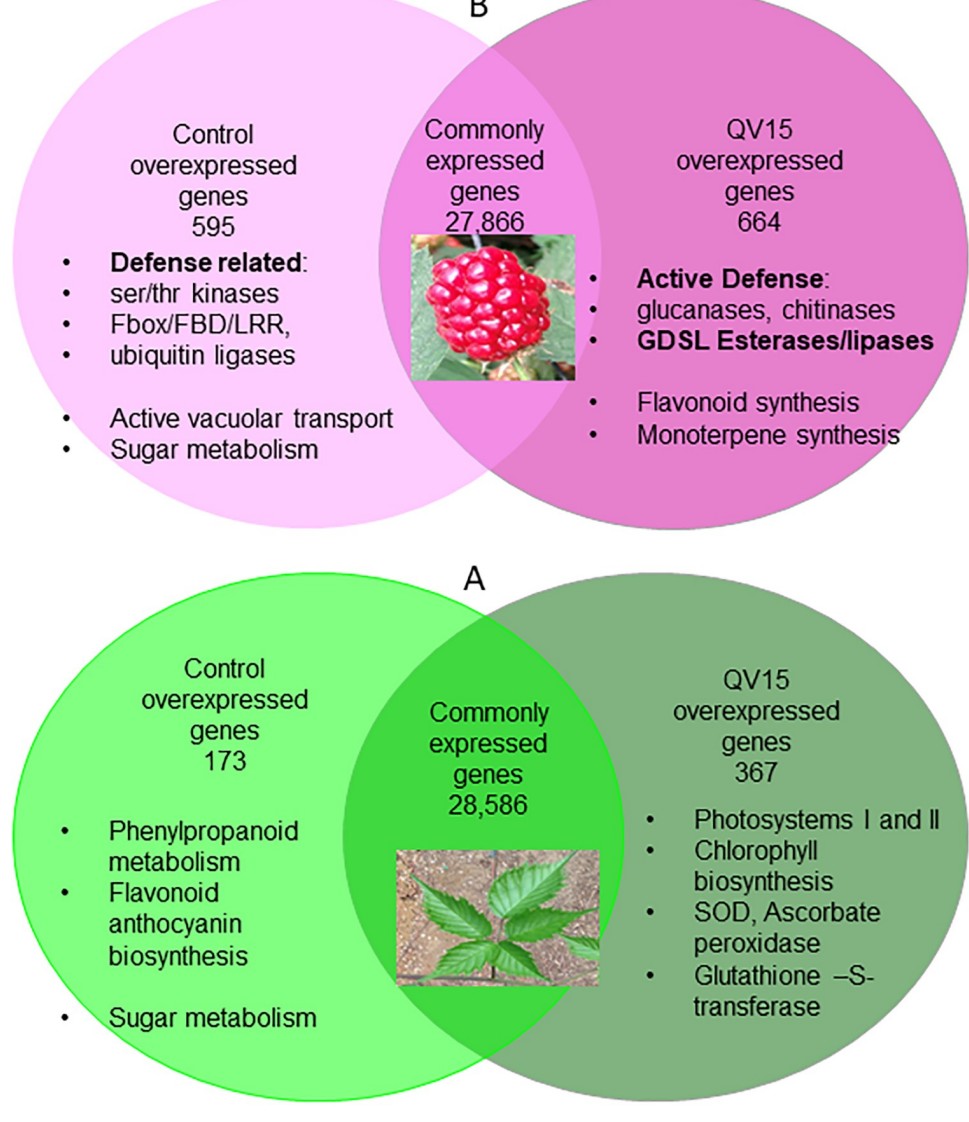

**Fig 3. Venn diagram of overexpressed and common genes in blackberry leaves (a) and fruits (b) from control and QV15-treated plants.**

## Phenolics and flavonoids characterization

Leaves of QV15 treated plants had lower values of total phenolics (-18%), total flavonoids (-33%) and total anthocyanins (-21%) than controls and differences were statistically significant. Total phenolic contents averaged 16.88 mg g$^{-1}$ and 13.69 mg g$^{-1}$ for control and QV15 treated plants, respectively. Total flavonols represent between 11 and 9% of total phenolics,

**Table 2. Photosynthetic pigments in blackberry leaves in controls and QV15 treated plants.**

| Samples | Chlorophyll A (mg g$^{-1}$) | Chlorophyll B (mg g$^{-1}$) | Carotenoids (mg g$^{-1}$) |
|---|---|---|---|
| Control | 0.57 ± 0.007 (a) | 0.23± 0.006 (a) | 0.38± 0.004 (a) |
| QV15 | 0.72± 0.004 (b) | 0.36± 0.017 (b) | 0.42± 0.002 (b) |

with 1.8561 mg g$^{-1}$ and 1.2463 mg g$^{-1}$ for controls and QV15 treated plants, respectively. Total anthocyanins represent around 1.6% of total phenolics, with 0.284 mg g$^{-1}$ and 0.2209 mg g$^{-1}$ for controls and QV15 treated plants, respectively (Table 3).

Quantification of total phenolics, flavonols and anthocyanins in blackberry leaves and fruits. Values are the average of 3 replicates ± SD. Different letters denote statistically significant differences between control and treated leaves (a,b) or fruits (x,y), according to LSD test (p<0.05).

Fruits showed significantly higher levels of phenols in controls (5.04 mg/g) than QV15-treated plants (4.41mg/g), while total flavonoids and anthocyanins showed non-significant differences (Table 3). Interestingly, all bioactives were more abundant in leaves except for anthocyanins; while total phenolics were 3-fold higher in leaves that fruits, flavonols were 6-fold higher.

Characterization of the methanolic extract of Blackberry leaves and fruits using UHPLC/ ESI-qTOF-MS provided a good separation profile (Fig 4). The visualization of both chromatograms profile, run of 25 min, revealed more intense and well-resolved chromatographic peaks in negative compared to the positive ion mode.

This method allowed separation of three groups of compounds: phenolic compounds eluted first, from min 0,5 to min 13; then, ursane-type triterpene saponins, from 12,5 to 18,5 minutes, and chlorophyll break down products from 18,5 to 22 min (Fig 4). Chlorophyll breakdown products were higher in controls than in QV15 treated plants. When fruits were analyzed, phenolic compounds eluted first; ursanes and chlorophyll breakdown products were not present (S1 Fig).

In Blackberry leaves, characteristic flavonols were kaempferol and quercetin derivatives, and (−)-epicatechin among catechols, being gallic acid the most abundant phenolic acid. Table 4 shows that the most abundant flavonols identified in leaves were quercetin-3-O-rutinoside, kempferol-O-glucoside, quercetin-3-O-glucoside and kaempferol-3-O-rutinoside, being quercetin 3-O-glucoside the less abundant and kaempferol-3-O-rutinoside the most abundant.

Identification and quantification of predominant compounds, expressed in µg/g, of phenolic compounds in leaf samples. Data is the average of 3 samples, with two injections each. <LoQ: below limit of quantitation (LoQ).

All of them except for kaempferol-O-rutinoside were higher in controls than in QV15-treated blackberries, as well as phenolic acids; interestingly, kempferol-O-rutinoside concentration was over 50% higher, and a marked decrease (54%) in (-)- epicatechin was observed (Table 4).

In Blackberry fruits, characteristic cathecols were epicatechin, catechin, and flavonols were represented by kaempferol and quercetin derivatives; vanillic acid was also present in relevant amounts (Table 5). Epicatechin was by far, the most abundant compound in red fruits, 200 µg/ g on average, while flavonols and anthocyanins were on the 5–10 µg/g range. The most abundant flavonols were quercetin 3-O-glucoside, quercetin-3-O-rutinoside, kaempferol-3-O-

**Table 3. Leaf and fruit bioactives in controls and QV15 treated plants.**

| Samples | Phenols (mg g$^{-1}$) | Flavonols (mg g$^{-1}$) | Anthocyanins (mg g$^{-1}$) |
|---|---|---|---|
| Control Leaves | 16.88 ± 0.292 (b) | 1.85 ± 0.061 (b) | 0.28 ± 0.007 (b) |
| QV15 Leaves | 13.69 ± 0.621 (a) | 1.24 ± 0.078 (a) | 0.22 ± 0.026 (a) |
| Control Fruit | 5.04 ± 0.074 (x) | 0.31 ± 0.012 (x) | 0.64 ± 0.015 (x) |
| QV15 Fruit | 4.41 ± 0.187 (y) | 0.29 ± 0.003 (x) | 0.68 ± 0.004 (x) |

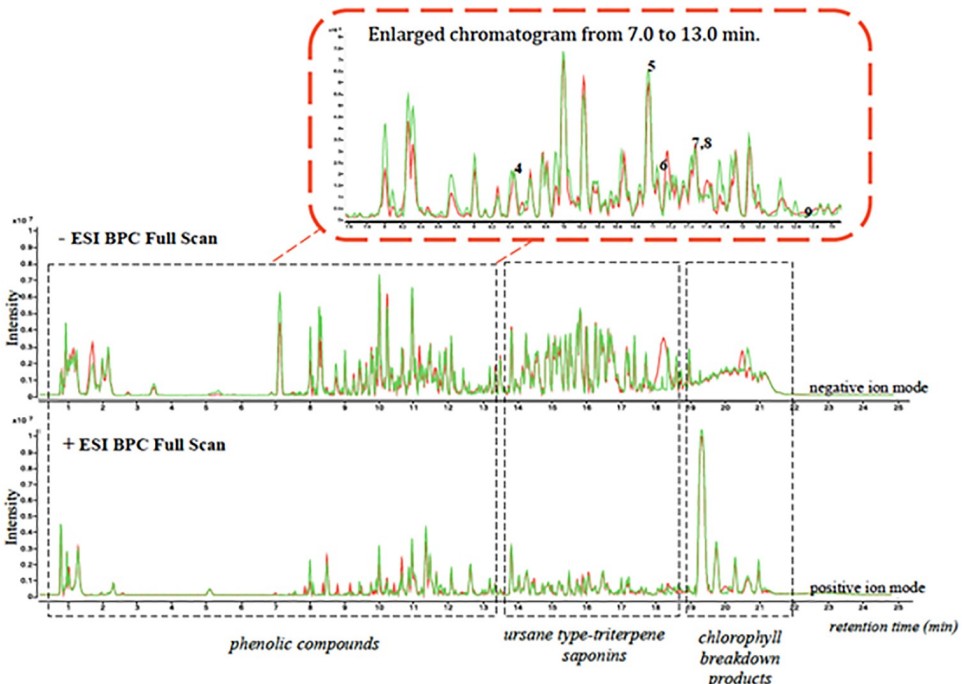

**Fig 4. Overlaid chromatograms (positive and negative ion mode) obtained from LC/MS/TOFF analysis of the methanolic extract of blackberry leaf samples.** Control samples are represented in Green while QV15 samples appear in red.

**Table 4. Identification and quantification of phenolic compounds in leaf samples of controls and QV15 treated plants.**

| Peak No. | Compounds | $t_R$ (min) | Molecular Formula | Monoisotopic Mass | m/z experimental[b] | Area average (control) | µg/g | Area average (QV15) | µg/g |
|---|---|---|---|---|---|---|---|---|---|
| 1 | gallic acid | 3.0 | $C_7H_6O_5$ | 170.0215 | [M-H]⁻ = 169.0149 | 3.70E+05 | 9.143 | 3.52E+05 | 8.237 |
| 2 | gentisic acid | 8.3 | $C_7H_6O_4$ | 154.0266 | [M-H]⁻ = 153.0196 | 1.14E+05 | 3.570 | 8.73E+04 | 2.740 |
| 3 | 6,7-dyhidroxycoumarin | 9.2 | $C_9H_6O_4$ | 178.0266 | [M-H]⁻ = 177.0181 | 8.51E+04 | <LoQ | 1.38E+05 | <LoQ |
| 4 | (-)-epicatechin | 9.4 | $C_{15}H_{14}O_6$ | 290.0790 | [M-H]⁻ = 289.0723 | 3.63E+06 | 6.793 | 7.18E+05* | 3.124 |
| 5 | quercetin-3-O-glucoside | 11.0 | $C_{21}H_{20}O_{12}$ | 464.0955 | [M-H]⁻ = 463.0887 | 6.41E+06 | 7.045 | 6.49E+06 | 7.324 |
| 6 | quercetin-3-O-rutinoside | 11.1 | $C_{27}H_{30}O_{16}$ | 610.1534 | [M-H]⁻ = 609.1494 | 1.08E+07 | 28.201 | 7.96E+06 | 17.827 |
| 7 | kaempferol-3-O-glucoside | 11.5 | $C_{21}H_{20}O_{11}$ | 448.1006 | [M-H]⁻ = 447.0938 | 4.23E+06 | 9.806 | 3.81E+06 | 8.148 |
| 8 | kaempferol-3-O-rutinoside | 11.5 | $C_7H_6O_3$ | 138.0317 | [M-H]⁻ = 593.1520 | 4.11E+06 | 37.056 | 5.70E+06 | 58.119 |
| 9 | luteolin | 12.7 | $C_{15}H_{10}O_6$ | 286.0477 | [M-H]⁻ = 285.0395 | 8,27E+05 | <LoQ | 1,04E+06 | <LoQ |

**Table 5. Identification and quantification of phenolic compounds in fruit samples of controls and QV15 treated plants.**

| N° | NAME COMPOUND | MW (g/mol) | RT (Q-TOF) | Chemical Formula | Monoisotopic Mass | Area average (control) | µg/g | Area average (QV15) | µg/g |
|---|---|---|---|---|---|---|---|---|---|
| 1 | Salicyclic acid | 138,12 | 11,5 | $C_7H_6O_3$ | 138,0317 | 1,00e+05 | <LOQ | 1,40e+05 | <LOQ |
| 2 | Vanillic acid | 168,15 | 9,2 | $C_8H_8O_4$ | 168,1467 | 6,14E+04 | 11 | 6,96e+04 | 14 |
| 3 | Chlorogenic acid | 354,31 | 8,9 | $C_{16}H_{18}O_9$ | 354,0951 | 6,71E+05 | <LoQ | 5,95E+05 | <LoQ |
| 4 | Phlorizin | 436,41 | 11,3 | $C_{21}H_{24}O_{10}$ | 436,1369 | 1,11E+05 | <LoQ | 4,98e+04 | <LoQ |
| 5 | (-)-epicatechin | 290,27 | 9,4 | $C_{15}H_{14}O_6$ | 290,0790 | 6,24E+07 | 231,197 | 5,79E+07 | 214,749 |
| 6 | (+)-catechin | 290,27 | 8,5 | $C_{15}H_{14}O_6$ | 290,0790 | 1,49E+06 | 4,914 | 1,26E+06 | 4,142 |
| 7 | Kaempferol-3-O-glucoside | 448,38 | 11,5 | $C_{21}H_{20}O_{11}$ | 448,1006 | 3,52E+05 | 0,761 | 4,00e+05 | 0,864 |
| 8 | Kaempferol-3-O-rutinoside | 594,52 | 11,5 | $C_{27}H_{30}O_{15}$ | 594,1585 | 3,66E+05 | 1,825 | 4,84e+05 | 2,415 |
| 9 | Quercetin | 302,24 | 12,4 | $C_{15}H_{10}O_7$ | 302,0459 | 6,35E+04 | <LoQ | 5,64e+04 | <LoQ |
| 10 | Quercetin-3-O-glucoside | 464,38 | 11,1 | $C_{21}H_{20}O_{12}$ | 464,0955 | 2,35E+06 | 5,016 | 2,32e+06 | 3,244 |
| 11 | Quercetin-3-O-rutinoside | 610,52 | 11,0 | $C_{27}H_{30}O_{16}$ | 610,1534 | 1,53E+06 | 3,104 | 1,52E+06 | 2,871 |
| 12 | Quercetin-3-O-galactoside | 464,3763 | 9,4 | $C_{21}H_{20}O_{12}$ | 464,0955 | 6,22E+05 | a | 7,08E+05 | a |
| 13 | Malvidin-3-O-galactoside | 493,39 | 9,6 | $C_{23}H_{25}O_{12}$ | 493,1346 | | <LoQ | | <LoQ |
| 14 | Delphinidin | 303,2436 | 9,1 | $C_{15}H_{11}O_7$ | 303,0505 | 1,18E+06 | a | 1,35E+06 | a |
| 15 | Cyanidin-3-O-arabinoside | 419,3589 | 9,4 | $C_{20}H_{19}O_{10}$ | 419,0978 | 3,52E+05 | a | 2,64E+05 | a |
| 16 | Cyanidin-3-O-glucoside | 448,3769 | 9,2 | $C_{21}H_{20}O_{11}$ | 448,1006 | 4,32E+07 | 2959,344 | 3,90E+07 | 2672,706 |

rutinoside, and kempferol-O-glucoside while the most relevant anthocyanins were cyanidin 3-O-glucoside and cyanidin 3-O-arabinoside; delphinidin was detected only in QV15 treated plants. All of them are described from higher to lower abundance. Controls showed higher concentrations of all compounds except for kempferol derivatives, as occurred in leaves.

Identification and quantification on phenolic compounds present in blackberry fruits of control and QV15 treated samples. Data is the average of 3 biological replicates, 2 injections each. a) no standard available for quantification. <LoQ: below limit of quantitation (LoQ).

In addition to those compounds, an exhaustive analysis of other peaks was carried out by comparing the full TOF mass spectral data features to a list of possible compounds showing that mass. Some interesting compounds with bioactive potential were identified such as procyanidins, or galactosyl-diacyl-glycerid derivatives and the ellagic tannin sanguiin H6, which could be identified only in the negative mode. Interestingly, all were higher in controls except for a galactosyldiacy-glycerid, which appeared only in QV15 treated fruits, in the positive mode, showing a large area in the chromatogram (S1 Fig).

## RT-qPCR analysis of core and regulatory genes of the flavonol-anthocyanin pathway

Figs 5 and 6 show differential expression of the regulatory and core genes of the flavonol-anthocyanin pathway in leaves and fruits, respectively. Control expression is marked as 1, therefore, expression values over one indicate overexpression in QV15 treated plants; conversely, values below one can be interpreted as overexpression in controls. Asterisks indicate statistically significant differences.

In general, expression of the flavonol-anthocyanin pathway core genes was higher in controls (Fig 5). Two isoforms were studied for *RuPAL*, *RuCHI*, and *RuGST*. Both *RuPAL* isoforms were overexpressed in control plants, being overexpression of *RuPAL1* significantly higher in controls than in QV15 treated plants. *RuCHI1*, *RuFLS*, *RuLAR*, *RuANR*, *RuDFR* and *RuANS* were overexpressed in control plants. Last enzyme of phenylpropanoids, *Ru4CL*, and last of early flavonol biosynthetic genes, *RuF3H*, were significantly overexpressed in QV15 treated plants. Genes encoding for the other enzymes *RuC4H*, *RuCHS*, *RuCHI2*, *RuF3´5´H* and *RuF3´H* were similarly expressed in control and QV15 treated plants. It was also found that *RuGST1* (glutathione S transferase 1) was overexpressed in control plants, while *RuGST2* (glutathione S transferase 2) was overexpressed in QV15 treated plants (S2 Table). In leaves of QV15-inoculated plants, the transcription factors *RuMYB3* and *RuMYB5* were significantly overexpressed (insert Fig 5), while *RuMYB1* and *RuMYB4* were overexpressed in controls.

In fruits, expression of the flavonol-anthocyanin pathway core genes was higher in controls (Fig 6). Two isoforms were studied for *RuPAL*, *RuCHI*, and *RuGST*. Both *RuPAL* isoforms were overexpressed in control plants, being overexpression of *RuPAL2* significantly higher in controls than in QV15 treated plants. *Ru4CL*, *RuCHI1*, *RuCHI2* (early genes of flavonol-anthocyanin pathway), *RuF3´5´H*, *RuF3´H*, *RuFLS* (late genes of flavonol-anthocyanin pathway), and *RuANS* were overexpressed in control plants. *RuCHS*, *RuF3H* (early steps), *RuLAR*, (catechin pathway), and *RuDFR* (anthocyanin pathway) were significantly overexpressed in QV15 treated plants. It was also found that *RuGST2* (glutathione S transferase 1) was overexpressed in control plants (values below 1), while *RuGST1* (glutathione S transferase 2) was overexpressed in fruits of QV15 treated plants (S2 Table). In QV15 inoculated plants, the

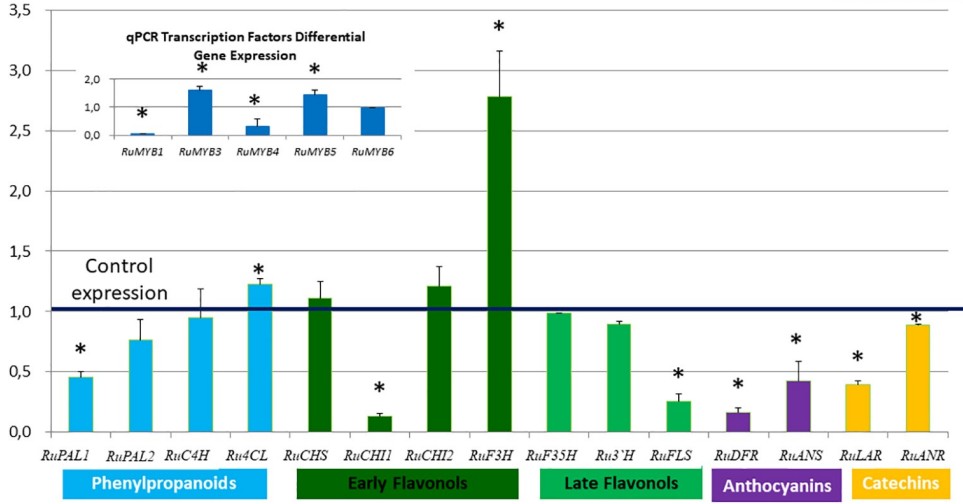

**Fig 5. Flavonol-anthocyanin pathway gene expression analyzed by RT-qPCR in leaves.** The line set at value of 1 represents gene expression in controls, so values over one indicate overexpression in QV15 treated plants and values below one indicate overexpression in controls. Phenylalanine ammonio-lyase (*RuPAL1* and *RuPAL2*), Cinammate 4 hydroxylase (*RuC4H*), 4-coumaryl-CoA ligase (*Ru4CL*), Chalcone synthase (*RuCHS*), Chalcone Isomerase1 (*RuCHI1*), Chalcone Isomerase2 (*RuCHI2*), Flavonol-3-hydroxylase (*RuF3H*), Flavonoid 3´5´hydroxylase (*RuF3´5´H*), Flavonoid 3´hydroxylase (*RuF3´H*), Flavonol synthase (*RuFLS*), Leucoanthocyanidin reductase (*RuLAR*), Anthocyanidin reductase (*RuANR*), Dehydroflavonol reductase (*RuDFR*), Anthocyanidin synthase (*RuANS*). Insert: Flavonol-anthocyanin pathway regulatory genes. Asterisks indicate significant differences, according to Fisher test ($p < 0.05$).

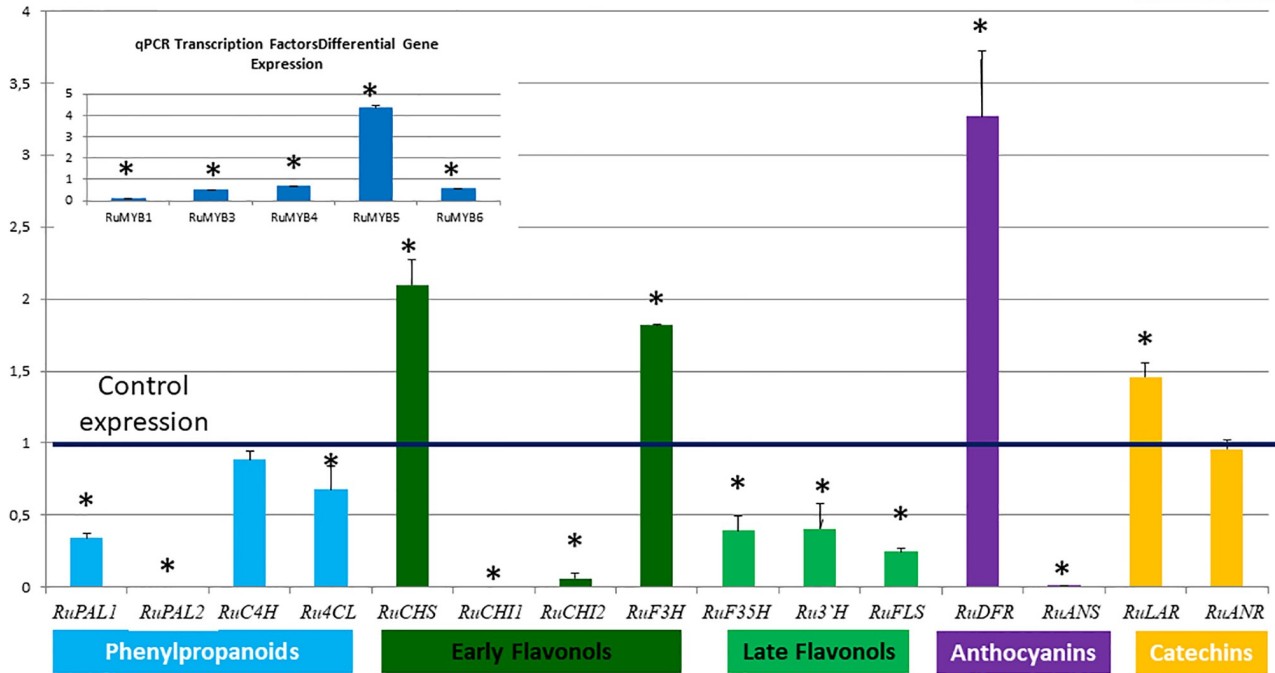

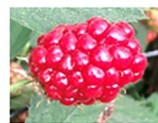

**Fig 6. Flavonol-anthocyanin pathway gene expression analyzed by RT-qPCR in fruits.** The line set at value of 1 represents gene expression in controls, so values over one indicate overexpression in QV15 treated plants and values below one indicate overexpression in controls Phenylalanine ammonio-lyase (*RuPAL1* and *RuPAL2*), Cinammate 4 hydroxylase (*RuC4H*), 4-coumaryl-CoA ligase (*Ru4CL*), Chalcone synthase (*RuCHS*), Chalcone Isomerase1 (*RuCHI1*), Chalcone Isomerase2 (*RuCHI2*), Flavonol-3-hydroxylase (*RuF3H*), Flavonoid 3´5´hydroxylase (*RuF3´5´H*), Flavonoid 3´hydroxylase (*RuF3´H*), Flavonol synthase (*RuFLS*), Leucoanthocyanidin reductase (*RuLAR*), Anthocyanidin reductase (*RuANR*), Dehydroflavonol reductase (*RuDFR*), Anthocyanidin synthase (*RuANS*). Insert: Flavonol-anthocyanin pathway regulatory genes. Asterisks indicate significant differences, according to Fisher test ($p < 0.05$).

transcription factor *RuMYB5* was significantly overexpressed, while *RuMYB1*, *RuMYB3*, *RuMYB4* and *RuMYB6* were overexpressed in controls (insert, Fig 6).

## Discussion

The results presented in this study indicate that QV15 triggers plant metabolism, improving plant fitness, adaptation to biotic stress and stimulating the flavonol-anthocyanin pathway in blackberry.

The responses triggered by this strain in the plant involves activation of gene expression related to photosynthesis and oxidative stress and related to specialized protective enzymes. The abundant transcripts related to photosynthesis found in leaves of QV15 treated plants reflect an active system for light reactions, an improvement in the efficiency of the photosynthetic electron transport chain, supported by overexpressed genes related to biosynthesis of photosynthetic pigments, mainly chlorophylls A and B. This expression is consistent with the significantly higher levels in chlorophylls and carotenoids of QV15-treated plants Table 2), also reported for other *Bacillus* strains [43]. Furthermore, the UHPLC/ESI-qTOF-MS analysis indicated lower levels of chlorophyll breakdown products in elicited plants, so the positive effects on pigments could be explained by either an increased biosynthesis, or a decreased

degradation, or both (Fig 4, Table 2). The high activity of light reactions seems to be coordinated with an active carbon fixation, as overexpressed transcripts of ribulose bisphosphate carboxylase (RuBisCO) are found. Consistent with the high activity of light reactions, abundant transcripts of the enzymatic pool of antioxidants were also observed and overexpressed in QV15 treated plants (supplementary material), suggesting a protective role against oxidative stress, and confirming enhanced plant fitness [44–45]. A striking overexpression of the isoenzyme glutathione-S-transferase (GST2), an enzyme with a strong protective role against oxidative stress, contributes to the enhanced plant fitness, as it is consistent with the high expression of the enzymatic pool of antioxidants. Furthermore, GST has been reported to be a molecular marker of induced resistance signaling mediated by ethylene in *A.thaliana* [46] and strongly related to phenylpropanoid-flavonoid transport within the plant [17].

Overexpressed genes in fruits, in one hand, reveal high vacuolar activity and an active sucrose metabolism, and in the other hand, the strong stress defense response and cell death is reflected in the many transcripts of ubiquitin-protein ligases, serin/theronin kinases and Fbox/FBD/LRRs [47] in controls, that in fact show a higher disease incidence. The F-box genes constitute one of the largest gene families in plants involved in degradation of cellular proteins. F-box proteins can recognize a wide array of substrates and regulate many important biological processes among which are biotic and abiotic stress responses. Conversely, in QV15-treated fruits, defense response relies on specialized defense enzymes such as subtilisin, glucanases and chitinases [48], and on a striking overexpression of GDSL esterase/lipases, a family of proteins that has been related to secondary metabolite synthesis and plant defense [42]. This reveals the different pathways involved in protection and highlights the systemic response in QV15 treated plants.

Stimulating the photosynthetic process suggests that the increase in the carbon fixed will be fed into growing leaves, flowers, and fruits, enhancing plant growth and probably increasing fruit yield, as reported for some beneficial bacterial strains [11–45]. This active metabolism provides a metabolic support to the high increase of flowers recorded at flowering was expected to be reflected into a fruit yield increase. However, no significant increases in fruit yield were detected probably due to the Mildew outbreak after flowering, in which QV15 treated plants showed less disease symptoms than controls, with a protection that ranged between 87 to 68% along plant cycle (Table 1) [49]. That protection involves deviation of plant resources to plant defense, therefore compromising plant yield, as balancing immunity and plant yield is key for survival [50].

Our rationale was to demonstrate the role of flavonoids in adaptation to biotic stress, with a double aim, i) protection and ii) fruit quality. On one hand, to stimulate flavonoid synthesis on leaves to improve plant defense, as these secondary metabolites have been reported to play a relevant role in defense, being of great importance against biotic stress [51–52]. On the other hand, to benefit from this stimulation to enrich fruits on flavonoids and anthocyanin contents [53] as they are bioactive molecules good to prevent onset of disease [54]. This strategy is in line with the rationale reported by Taye-Desta et al [54] who approaches changes on flavonoid metabolism on pathogen infected plants, with a similar aim. More precisely, the flavonoids reported here, refer to the profile of total phenolics, flavonols and anthocyanins.

Flavonoids may alternatively play a role as phytoalexins or phytoanticipins, depending on the plant species, or even within different tissues of the same plant [25–55], as has been reported for sakuranetin, a common flavonoid in the Poaceae family [56]. An increase upon pathogen challenge would indicate a role as phytoalexin [57], while a decrease upon pathogen challenge would indicate a phytoanticipin role [24]. Moreover, the aglycons of flavonols have been reported to be more effective against fungi than their methyl derivatives [58–59] while flavanes, proanthocyanidins and isoflavones have been reported to be more effective against

bacteria [60]. The relevance of the topic shows in the increasing number of reports about different flavonols and their role in different plant species reviewed in Taya-Desta et al. [55]. In control plants, flavonols were higher than in QV15 treated plants; this situation would indicate a role as phytoalexins in blackberry leaves, as flavonols increase in response to pathogen elicitors, and one of the criteria to qualify for phytoalexin is "to accumulate upon pathogen infection" [55].

However, despite the lower flavonol concentration found in leaves of QV15 treated plants as well as that of total phenolics (Table 3), there was lower disease incidence. Consistent with the role of phytoanticipins, flavonols would be effectively transformed into another molecule, the phytoalexin, also resulting in lower flavonol levels in plant [55]. This statement is supported in part by the striking lower concentration in (-)-epicatechin and quercetin derivatives registered in QV15 treated leaves (Table 4) and fruits (Table 5), and overexpression of key genes in the pathway (Figs 5 and 6). Interestingly, a noticeable accumulation of kempferol-3-rutinoside was detected (Tables 4 and 5) only in QV15 treated plants, suggesting a putative role in defense which is worth exploring since differential effects of each type of flavonol have been reported [61, 62]. Irrespective of the fate of each molecule, the net balance of flavonol pool results in lower concentration in QV15 treated plants, which still remain more protected. This higher protection is probably connected to the metabolic reprogramming induced by QV15, shown in the enhanced specialized defense based on enzymes revealed by transcriptome analysis (Fig 3). As regards to fruits, no differences were found between controls and QV15-treated fruits in neither bioactive concentration, and still, Kempferol-3-rutinoside was strikingly high as in leaves, reinforcing the notion of a relevant role of this molecule in plant adaptative response.

Transcript profiling revealed coordinated increased transcript abundance for genes encoding enzymes of committing steps in the flavonol-anthocyanin pathway as well as in the regulators in QV15 treated plants, which was different in leaves and fruits. In leaves, only *Ru4CL* and *RuF3H*, the last enzymes in the phenylpropanoid pathway (*Ru4CL*), and last in the early flavonol-anthocyanin pathway (*RuF3H*), respectively, were overexpressed, suggesting a pivotal role for RuF3H in the control of the flavonol-anthocyanin pathway, consistent with the before mentioned metabolomic changes. The overexpression of key genes in the pathways ensures the carbon flux to that metabolic cluster, as enzymes involved in this pathway have been reported to cluster associated to the ER membrane for a better performance [63]. In fruits, overexpression of first and last gene of the early flavonol biosynthetic genes, first of anthocyanins and first for catechins revealed an active anthocyanin biosynthesis in QV15 fruits, anticipating the massive biosynthesis that is about to occur upon complete fruit maturation [53]. As flavonol concentration is significantly lower in leaves of QV15 plants and higher anthocyanin concentration in fruits, together with a high abundance of GST transcripts, we hypotesize that leaf flavonols are being actively translocated to support anthocyanin synthesis in fruits [16]. This process was more effective under the influence of QV15. Interestingly, the homologous to the positive regulators of late steps in the flavonol-anthocyanin pathway, *RuMYB3* and *RuMYB5* [64] were overexpressed in leaves, and only *RuMYB5* was in fruits, reinforcing the hypothesis of flavonoids being actively formed in leaves while leaf anthocyanin synthesis is inhibited and is activated in fruits. This suggests that *RuMYB3* could behave as anthocyanin repressor in blackberry, as the mode of control of the flavonoid pathway is quite specific of the species and moment of development [64]; furthermore, *RuMYB5* appears as the target for biotic stress adaptation used by this *Bacillus* strain.

Finally, the better performance in the inoculated plants against the pathogen could rely in other molecules, leaving a partial role in defense for flavonols, so the decrease would be due to translocation to fruits, to fulfill sink demand for anthocyanin biosynthesis, as they are vastly

produced in leaves of blackberry plants [3]. Consistent with the partial role of flavonols in defense, the untargeted metabolomic profile revealed a characteristic series of triterpenoid pentacyclic saponins specific to the *Rubus* genus, ursanolic acids [41], which have been attributed to have an antimicrobial potential [64]; these compounds were more abundant in QV15 treated plants.

In summary, elicitation with QV15 has revealed a pivotal role of RuF3H controlling carbon fluxes towards the different sinks in the flavonol-anthocyanin pathway in blackberry and a relevant action of RuMYB5 in its control. The abundance of Kempferol-3-*O*-rutinoside leaves an open question about its role in defense. This C demand is supported by an activation of the photosynthetic machinery and boosted by a coordinated control of ROS into a sub-lethal range, in which GST2 seems to have a strong participation, and results in enhanced protection to biotic stress.

## Supporting information

**S1 Fig. Blackberry fruits chromatograms.** Overlaid Chromatograms (positive and negative ion mode) obtained from LC/MS/TOFF analysis of the methanolic extract of BlackBerry fruit samples. Control samples are represented in green while QV15 samples appear in red.
(TIF)

**S1 Table. Number of mappable samples and paired readings per sample.**
(DOCX)

**S2 Table. Expression of glutathione S transferases.** Glutathione S transferases gene expression analyzed by RT-qPCR in leaves and fruit. Asterisks indicate significant differences, according to Fisher test ($p < 0.05$).
(DOCX)

**S3 Table. Primers designed to RT-qPCR expression analysis.**
(DOCX)

**S1 File. Differentially expressed genes in leaves.**
(XLSX)

**S2 File. Differentially expressed genes in fruits.**
(XLSX)

**S3 File.**
(XLSX)

**S4 File.**
(XLSX)

## Acknowledgments

Authors thank for Agricola El Bosque S.L. "La Canastita" for providing help in blackberry crop.

## Author Contributions

**Conceptualization:** F. Javier Gutierrez-Mañero, Beatriz Ramos-Solano.

**Data curation:** Enrique Gutiérrez-Albanchez, Ana García-Villaraco, Beatriz Ramos-Solano.

**Formal analysis:** Enrique Gutiérrez-Albanchez, Ana Gradillas, Antonia García, Ana García-Villaraco.

**Funding acquisition:** Antonia García, F. Javier Gutierrez-Mañero, Beatriz Ramos-Solano.

**Investigation:** Enrique Gutiérrez-Albanchez, Ana Gradillas, Ana García-Villaraco, Beatriz Ramos-Solano.

**Writing – original draft:** Enrique Gutiérrez-Albanchez, Ana Gradillas.

**Writing – review & editing:** F. Javier Gutierrez-Mañero, Beatriz Ramos-Solano.

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
