## [Decision Letter · Decision Letter 0]

13 Feb 2020

PONE-D-19-31790

Elicitation with Bacillus QV15 reveals a pivotal role of F3H on flavonoid metabolism improving adaptation to biotic stress in blackberry

PLOS ONE

Dear Dr Ramos-Solano,

Thank you for submitting your manuscript to PLOS ONE. After careful consideration, we feel that it has merit but does not fully meet PLOS ONE’s publication criteria as it currently stands. Therefore, we invite you to submit a revised version of the manuscript that addresses the points raised during the review process.

We would appreciate receiving your revised manuscript by Mar 29 2020 11:59PM. To enhance the reproducibility of your results, we recommend that if applicable you deposit your laboratory protocols in protocols.io, where a protocol can be assigned its own identifier (DOI) such that it can be cited independently in the future. For instructions see: http://journals.plos.org/plosone/s/submission-guidelines#loc-laboratory-protocols

We look forward to receiving your revised manuscript.

Kind regards,

Anil Kumar Singh, Ph.D.

Academic Editor

PLOS ONE

Journal Requirements:

2. Please upload a new copy of Figure 2 as the detail is not clear. Please follow the link for more information: http://blogs.PLOS.org/everyone/2011/05/10/how-to-check-your-manuscript-image-quality-in-editorial-manager/

Reviewers' comments:

Reviewer's Responses to Questions

**Comments to the Author**

1. Is the manuscript technically sound, and do the data support the conclusions?

Reviewer #1: Yes

Reviewer #2: No

2. Has the statistical analysis been performed appropriately and rigorously? 

Reviewer #1: Yes

Reviewer #2: N/A

3. Have the authors made all data underlying the findings in their manuscript fully available?

Reviewer #1: No

Reviewer #2: No

4. Is the manuscript presented in an intelligible fashion and written in standard English?

Reviewer #1: Yes

Reviewer #2: No

5. Review Comments to the Author

Reviewer #1: The manuscript reports on the blackberry adaptation to biotic stress (mildew) due to elicitation by Bacillus amyloliquifaciens via overexpression of flavonoid metabolism genes. The article could be of interest to readers working in the area of beneficial plant-microbe interaction. Authors should address all the concerns raised in the manuscript pdf file highlighted with YELLOW.

Reviewer #2: The manuscript “Elicitation with Bacillus QV15 reveals a pivotal role of F3H on flavonoid metabolism improving adaptation to biotic stress in blackberry” in its present format is not fit for the publication.

• Using transcriptomics and qRT PCR analysis, many genes related to flavonoid metabolism have been examined of which RuF3H, was significantly overexpressed in QV15469 treated plants. The authors also claimed that RuF3H could be useful for biotic stress. To claim this, some biotic stress challenge experiments need to be carried out. What is the explanation for other genes, which were downregulated in QV15469 treated plants?

• The heat map provided in the manuscript is not clearly visible.

• What is the explanation of the low level of phenolic compounds in QV15 treated plants than control? Although one citation is there, (Taye-Desta K et al. Front Nat Prod Chem. 2016; 2:3-49), but not sufficiently discussed.

• Except introduction part, all parts need to be improved. Especially materials and methods section is crappy.

• The references given in the main text of the manuscript must be corrected, such as instead of [19, 20, 21, 22, 23] it could be [19-23].

• Proper “prime” sign should be used. (Line 96-97 and also in other parts; (RuCHI2), Flavonol-3-hydroxylase (RuF3H), Flavonoid 3´5´hydroxylase (RuF3´5´H), Flavonoid 3´hydroxylase (RuF3’H),)

• The figures mentioned in the main text need to be mentioned in a proper way, Somewhere, mentioned as Fig, in some places figure etc.

• The raw RNA-sequenced data must be submitted online in appropriate database and should also be mentioned in the MS.

6. PLOS authors have the option to publish the peer review history of their article (what does this mean?). If published, this will include your full peer review and any attached files.

Reviewer #1: Yes: Charu Lata

Reviewer #2: No

---

## [Author Response · Author response to Decision Letter 0]

25 Mar 2020

Reviewer #1: The manuscript reports on the blackberry adaptation to biotic stress (mildew) due to elicitation by Bacillus amyloliquifaciens via overexpression of flavonoid metabolism genes. The article could be of interest to readers working in the area of beneficial plant-microbe interaction. Authors should address all the concerns raised in the manuscript pdf file highlighted with YELLOW.

Dear Reviewer. Thank you very much for your comment. However, the ms. has no yellow marks. We understand that we need to receive the marked version of the ms. we have requested so to the PLOS central office and look forward to receive it soon.

Comments received on march 9th. 

There are few major concerns which the authors need to address in this paper before it could be finally accepted for publication:

1. No details of raw/Fastq sequence submission mentioned in the manuscript;

Raw data was provided in supplementary material although the file was named as “S1_file. differentially expressed genes in leaves” and “S2_file. differentially expressed genes in fruits”. It has now been separated into different supplementary files as follows: S1_ raw data for leaves, S2_raw data for fruits, S3_ differentially expressed genes in leaves, S4_differentially expressed genes in fruits.

2. A correlation analysis between the RNASeq data and qRT-PCR data is a must for establishing the validity of the results obtained;

As we understand, you are requesting a correlation between gene expression obtained by RTqPCR and RNAseq. To our knowledge, this analysis is usually requested when conclusions are drawned from RNAseq analysis only. A number of studies have been carried out in the first moments of RNAseq boom to establish this relationship and validity, aiming to avoid doing both analysis. 

However, our rationale in this study was to use RNAseq and qPCR with absolutely different objectives. RNAseq aimed to get an overview of the plant status at a given point (peak in fruiting) to gain knowledge on plant reprogramming (leaves and fruits) after several doses of our strain delivered through the roots along the growth period. This analysis has revealed that plants under the influence of QV15 show a completely different pattern in gene expression, so QV15 induces an specific reprogramming (356 upregulated genes) of the plant that is further highlighted upon the mildew outbreak. As a matter of fact, this reprogramming reveals that photosynthesis related genes (54 transcipts upregulated) are affected by QV15, while only 7 dealing with secondary metabolism, and more precisely to terpenes are upregulated. On the other hand, controls under the influence of mildew only, show a reprogramming (173 upregulated genes) that mainly affects secondary metabolism (21 transcripts upregulated), most of which are related to the flavonol-anthocyanin pathway. So the RNAseq has revealed the different reprogramming in QV15 and control plants, and this was included in the discussion.

On the other hand, RT qPCRs aims to study changes on the flavonol-anthocyanin pathway in both organs. The study is done with primers that have been specifically designed for Rubus var Loch Ness, so an excellent precission is achieved. The differential expression is consistent with the situation revealed with RNAseq as the higher expression of the flavonol-anthocyanin pathway in controls is revealed, both in leaves and in fruits, with the exception of 4CL and F3H, in leaves, and CHS, F3H, DFR and LAR in fruits. Interestingly, this data is refered to a reference gene (actin) that has a constant expression, while this correction does not occur in RNAseq

In order to fulfil the reviewer’s request, we have done the correlation analysis for the genes of interest studied by RTqPCR. We found the following correlations when comparing the expression registered by RNAseq to the expression registered by RT qPCR. 

note C RNAseq/RT qPCR Qv15 RNAseq/RT qPCR

LEAVES

Similar to 4CL1: 4-coumarate--CoA ligase 1 (Nicotiana tabacum) 0,88131837 0,961424086

Similar to FLS: Flavonol synthase/flavanone 3-hydroxylase (Citrus unshiu)

 -0,837608 -0,88705718

FRUITS

Similar to PAL1: Phenylalanine ammonia-lyase 1 (Rubus idaeus) -0,00785298 0,92431717

Similar to FL: Flavonol synthase/flavanone 3-hydroxylase (Petunia hybrida) 0,95504953 0,9874591

Similar to LAR: Leucoanthocyanidin reductase (Desmodium uncinatum) -0,08856371 0,99093903

Similar to dfrA: Putative dihydroflavonol-4-reductase (Synechocystis sp. (strain PCC 6803 / Kazusa)) -0,82308285 -0,8937853

I would like to highlight that the plant species we are working with is not sequenced, so after alignement of sequences to the Rubus occidentalis genome, annotation of transcripts is made by comparing with all databases of all species, as annotations indicate. So we are comparing a precise qPCR analysis designed for this species to non-specific information including different species. For instance for F3H, we find different annotations for fruits (Similar to FL: Flavonol synthase/flavanone 3-hydroxylase (Petunia hybrida)) and for leaves (Similar to FLS: Flavonol synthase/flavanone 3-hydroxylase (Citrus unshiu)), in the same plant and same RNAseq analysis, so it seems very difficult to find a correlation; moreover, this annotation just mentioned has been selected among 4 different annotations for F3H which show worse correlations. Furthermore, many of these enzymes are known to have different isoenzymes depending on the organ and developmental stage, as widely known for PAL.

In our opinion, the requested correlation should not be included to confirm expression data for two reasons:

1) RNAseq was designed to detect general changes on gene expression following the biotic stress

2) Information about genes obtained by RNAseq is not precise as global database of similar sequences from different species is used

3) Our conclusions on the flavonol-anthocyanin pathway are based on the RT qPCR analysis which is more precise than the rough RNAseq data. 

3. Functional annotation using Gene Ontology should also be done 

Functional annotation with Gene Ontology has been done and is included in files S3 (column Q) and S4 (columna Q) for leaves and fruits, respectively. We will be glad to provide additional analysis if neccessary. Following a suggestion of the other reviewer, the corresponding section of the material and methods has been revised and this information has been included, as it was missing in the original version. We would like to apologize for this mistake.

4.A MAPMAN visualization of metabolic pathway may also be considered. 

We would like to thank the reviewer for this valuable suggestion. Mapman looks as a great tool for this purpose and we were working on it to get the software installed to provide a new figure for our data. Due to the limitations of software instalation in our university associated to coronavirus crisis, we are currently unable to provide this graph 

 

Dear Reviewer #2: We would like to thank you very much for your constructive criticism. However, we regret to hear that our ms is not ready for publication in its present form but we hope it will qualify after the requested modifications. All comments have been addressed in the order raised by you. 

• Using transcriptomics and qRT PCR analysis, many genes related to flavonoid metabolism have been examined of which RuF3H, was significantly overexpressed in QV15469 treated plants. The authors also claimed that RuF3H could be useful for biotic stress. To claim this, some biotic stress challenge experiments need to be carried out. 

As regards to the claim for biotic stress protection, we report here that there is a protection ranging between 87 to 68% in field conditions (table 1). Unfortunately, such an experiment cannot be repeated in production greenhouses since it is forbidden to release pathogens in the field. However, we have conducted the typical biotic stress protection experiment in Arabidopsis under controlled conditions with excellent results in different experiments carried on different years (published in 2008 and 2018). Our fist description to the ability of this bacillus strain to protect against biotic stress (Pseudomonas syringae DC300) and abiotic stress (NaCl) in Arabidopsis thaliana as compared to controls reported a protection of 60% and 70%, respectively (Barriuso et al 2008). Our next report has been recently published and establishes the similar mechanisms triggered by this strain in plant protection against biotic stress in arabidopsis (P.syringae DC300) and blackberry (mildew outbreak) (Gutierrez-Albanchez et al, 2018. Priming fingerprint induced by Bacillus amyloliquefaciens QV15, a common pattern in Arabidopsis thaliana and in field-grown blackberry https://doi.org/10.1080/17429145.2018.1484187). The response triggered by the bacterial strain in both plant species shares a decrease in ROS scavenging enzymes activity before and after pathogen challenge, an enormous increase in glucanase and chitinase activity after pathogen challenge and overexpression of PR1 after pathogen challenge, confirming the priming status induced by QV15 that results in effective protection. This reference is included in the text (ref 48) and provides support to our statement of the ability of this strain to trigger protection against biotic stress (line 556). The best scenario would be to repeat the field experiment but there is no way that a pathogen can be released in a production area as it would increase production costs due to phytochemical use and compromise fruit yield in the non-experimental area. 

Of course, expression of F3H and flavonol profiles in arabidopsis should be specifically studied in pathogen challenged plants treated with QV15 for a definitive demonstration of the involvement of this enzyme on plant protection. However, this is a new approach that we plan to address soon, in order to explore the main questions that arise from the study presented in this work: the role of F3H, MYB5, and K-3-glucoside in plant protection.

• What is the explanation for other genes, which were downregulated in QV15469 treated plants?

Plants live in a non-sterile environment and have to overcome any and many challenges that come along during their life. As they are sessile, the established mechanisms of protection are based on chemical molecules. It is widely accepted that upon pathogen challenge, plants undergo transcriptional and metabolic reprogramming involving synthesis of metabolites to fight invassion and survive (Mauch-Manni et al. 2017. Mauch-Mani B, Baccelli I, Luna E, Flors V. 2017. Defense priming: An adaptive part of induced resistance. In: S. S. Merchant, editor.Annual review of plant biology, Vol 68. Palo Alto: Annual Reviews;p. 485–512; Taye Desta et al 2016). This reprogramming is specific of each plant- pathogen interaction, and although changes in many plant-pathogen system have been studied and agreed on, when any of the elements of this pair change (plant or pathogen), changes are different to some extent. The same holds true for studies with elicitors/beneficial microorganisms in different plant species with even greater differences; these are greatly increased when the experimental work is done in field conditions. 

Getting to the question raised by the reviewer, we are also really curious about this downregulation and can only hypotesize some possible explanations that need additional experimentation to proof. This is supported on the priming effect also: reprogramming of metabolism along the growth period involves activation of other defense systems different to flavonols, so expression of flavonol-anthocyanin pathway genes is not increased upon pathogen challenge in QV15 treated plants.

Our current Research is on that line, trying to uravel the reprogramming that takes place in the plant when QV15 is delivered through the roots to induce a systemic response. We will be happy to provide additional information on this line sometime soon.

• The heat map provided in the manuscript is not clearly visible.

The figure of the heat map has been replaced by a higher quality one. Apologies for the inconvenience.

• What is the explanation of the low level of phenolic compounds in QV15 treated plants than control? Although one citation is there, (Taye-Desta K et al. Front Nat Prod Chem. 2016; 2:3-49), but not sufficiently discussed.

(L568-582 of the original version) The decrease on phenolic compounds detected in QV15-treated leaves and fruits is a fact. We think that controls show higher level of phenolic compounds because they are in an aggressive hypersensitive response against the fungal invasion progressm and flavonols are behaving as phytoalexins. Conversely, QV15-treated plants are more relaxed, as they have been primed for a long time and are therefore more prepared to fight the fungal invasion with other weapons, be it i) phytochemicals different to phenols like the triterpenes, or of other chemical nature and therefore, out of our analysis, or ii) be it enzymes, like chitinases or glucanases (48. Gutierrez-Albanchez et al 2018 https://doi.org/10.1080/17429145.2018.1484187). 

Discussion has been enriched discussing Taye-Desta review at some points to explain state of the arts in other species. 

• Except introduction part, all parts need to be improved. Especially materials and methods section is crappy.

The text has been revised and hopefully improved. We have paid special attention to the RNAseq description which we agree was not the best. However, we will be pleased to make further changes to more precise suggestions.

• The references given in the main text of the manuscript must be corrected, such as instead of [19, 20, 21, 22, 23] it could be [19-23].

The following modifications have been made in the introduction: L 39, L50, L73. 

• Proper “prime” sign should be used. (Line 96-97 and also in other parts; (RuCHI2), Flavonol-3-hydroxylase (RuF3H), Flavonoid 3´5´hydroxylase (RuF3´5´H), Flavonoid 3´hydroxylase (RuF3’H),)

L98, L482, L507, L866, L893. The proper prime sign has been included, specially for RuF3’H that now appears as RuF3´H

• The figures mentioned in the main text need to be mentioned in a proper way, Somewhere, mentioned as Fig, in some places figure etc.

The text has been revised and “figure” in the text has been replaced by fig, as indicated in the formatting instructions. A total of 14 replacements have been made.

• The raw RNA-sequenced data must be submitted online in appropriate database and should also be mentioned in the MS.

Raw data has been provided in supplementary material although the file was named as “S1_file. differentially expressed genes in leaves” and “S2_file. differentially expressed genes in fruits”. It has now been separated into different supplementary files as S1_ raw data for leaves, S2_raw data for fruits, S3_ differentially expressed genes in leaves, S4_differentially expressed genes in fruits. 

The reference to this files has been indicated in the ms. (lines 332-333 or the revised ms. With tracked changes)

---

## [Decision Letter · Decision Letter 1]

20 Apr 2020

Elicitation with Bacillus QV15 reveals a pivotal role of F3H on flavonoid metabolism improving adaptation to biotic stress in blackberry

PONE-D-19-31790R1

Dear Dr. Ramos-Solano,

We are pleased to inform you that your manuscript has been judged scientifically suitable for publication and will be formally accepted for publication once it complies with all outstanding technical requirements.

With kind regards,

Anil Kumar Singh, Ph.D.

Academic Editor

PLOS ONE

Additional Editor Comments (optional):

Reviewers' comments:

Reviewer's Responses to Questions

**Comments to the Author**

1. If the authors have adequately addressed your comments raised in a previous round of review and you feel that this manuscript is now acceptable for publication, you may indicate that here to bypass the “Comments to the Author” section, enter your conflict of interest statement in the “Confidential to Editor” section, and submit your "Accept" recommendation.

Reviewer #1: All comments have been addressed

2. Is the manuscript technically sound, and do the data support the conclusions?

Reviewer #1: (No Response)

3. Has the statistical analysis been performed appropriately and rigorously? 

Reviewer #1: (No Response)

4. Have the authors made all data underlying the findings in their manuscript fully available?

Reviewer #1: (No Response)

5. Is the manuscript presented in an intelligible fashion and written in standard English?

Reviewer #1: (No Response)

6. Review Comments to the Author

Reviewer #1: (No Response)

7. PLOS authors have the option to publish the peer review history of their article (what does this mean?). If published, this will include your full peer review and any attached files.

Reviewer #1: Yes: Charu Lata

---

## [Editor Report · Acceptance letter]

22 Apr 2020

PONE-D-19-31790R1 

Elicitation with Bacillus QV15 reveals a pivotal role of F3H on flavonoid metabolism improving adaptation to biotic stress in blackberry 

Dear Dr. Ramos-Solano:

I am pleased to inform you that your manuscript has been deemed suitable for publication in PLOS ONE. Congratulations! Your manuscript is now with our production department. 

With kind regards,

on behalf of

Dr. Anil Kumar Singh 

Academic Editor

PLOS ONE